# Safety Profile and Outcomes of Early COVID-19 Treatments in Immunocompromised Patients: A Single-Centre Cohort Study

**DOI:** 10.3390/biomedicines10082002

**Published:** 2022-08-18

**Authors:** Simona Biscarini, Simone Villa, Camilla Genovese, Mara Tomasello, Anna Tonizzo, Marco Fava, Nathalie Iannotti, Matteo Bolis, Bianca Mariani, Antonia Grazia Valzano, Letizia Corinna Morlacchi, Francesca Donato, Giuseppe Castellano, Ramona Cassin, Maria Carrabba, Antonio Muscatello, Andrea Gori, Alessandra Bandera, Andrea Lombardi

**Affiliations:** 1Infectious Diseases Unit, Foundation IRCCS Ca’ Granda Ospedale Maggiore Policlinico, 20122 Milan, Italy; 2Centre for Multidisciplinary Research in Health Science (MACH), University of Milano, 20122 Milano, Italy; 3Department of Pathophysiology and Transplantation, University of Milan, 20122 Milan, Italy; 4Clinical Laboratory, Foundation IRCCS Ca’ Granda Ospedale Maggiore Policlinico, 20122 Milan, Italy; 5Respiratory Unit and Cystic Fibrosis Adult Center, Foundation IRCCS Ca’ Granda Ospedale Maggiore Policlinico, 20122 Milan, Italy; 6A.M. & A. Migliavacca Center for Liver Disease, Division of Gastroenterology and Hepatology, Foundation IRCCS Ca’ Granda Ospedale Maggiore, 20122 Milan, Italy; 7Department of Nephrology, Dialysis, and Renal Transplantation, Foundation IRCCS Ca’ Granda Ospedale Maggiore Policlinico, 20122 Milan, Italy; 8Hematology Unit, Foundation IRCCS Ca’ Granda Ospedale Maggiore Policlinico, 20122 Milan, Italy; 9Department of Internal Medicine, Adult Primary Immunodeficiencies Centre, Foundation IRCCS Ca’ Granda Ospedale Maggiore, 20122 Milan, Italy

**Keywords:** monoclonal antibodies, sotrovimab, remdesivir, early treatments, COVID-19

## Abstract

Background: Early treatment with remdesivir (RMD) or monoclonal antibodies (mAbs) could be a valuable tool in patients at risk of severe COVID-19 with unsatisfactory responses to vaccination. We aim to assess the safety and clinical outcomes of these treatments among immunocompromised subjects. Methods: We retrospectively reviewed all nonhospitalized patients who received an early treatment with RMD or mAbs for COVID-19, from 25 November 2021 to 25 January 2022, in a large tertiary hospital. Outcomes included frequency of adverse drug reaction (ADR), duration of symptoms and molecular swab positivity, emergency department access, hospital or intensive care unit admission, and mortality in the 14 days following treatment administration. Results: Early treatments were administered to 143 patients, 106/143 (74.1%) immunocompromised, including 41 solid organ and 6 hematopoietic stem cell transplant recipients. Overall, 23/143 (16.1%) subjects reported ADRs. Median time from treatment start to SARS-CoV-2 nasopharyngeal swab negativity and symptom resolution was 10 (IQR 6–16) and 2.5 days (IQR 1.0–6.0), respectively, without differences between immunocompromised and nonimmunocompromised patients. In the 14 days after treatment administration, 5/143 patients (3.5%) were hospitalized and one died as a result of causes related to COVID-19, all of them were immunocompromised. Conclusions: RMD and mAbs have minimal ADRs and favourable outcomes in immunocompromised patients.

## 1. Introduction

More than two years have elapsed since the appearance of the severe acute respiratory syndrome coronavirus 2 (SARS-CoV-2), which is responsible for the coronavirus disease 2019 (COVID-19), yet therapeutic tools remain limited. Vaccination proved to be effective in preventing COVID-19 with varying results depending on the type of vaccine administered and/or by the SARS-CoV-2 variant involved [1,2,3]. This applies also to immunocompromised patients (e.g., primary immunodeficiencies, solid organ transplant (SOT) recipients, hematopoietic stem cell transplant (HSCT) recipients, oncologic patients, and those receiving immunosuppressive drugs for autoimmune conditions), although both efficacy and duration of immune protection are lower than in immunocompetent patients [4,5,6,7,8,9].

A single administration of monoclonal antibodies (mAbs) and a 3-day course of the antiviral remdesivir (RMD) are among the treatments administered to vulnerable patients early after infection, with the aim of halting the progression of COVID-19 toward more severe manifestations and outcomes. Data about the safety and effectiveness of these treatments in real-world scenarios are still scarce. This is especially true of immunocompromised patients, one of the groups of patients that can benefit more from these therapeutic approaches. This population was explicitly excluded from many of the randomized controlled trials conducted so far involving these drugs, and the majority of the participants had other risk factors, namely obesity, diabetes, an age of 55 years or older, and cardiac disease [10,11]. Recently, an interventional open-label study examining the safety of sotrovimab in immunocompromised patients with impaired humoral immunity against SARS-CoV-2 has been registered on the ClinicalTrial.gov platform [12] and can help to shed light on this issue. However, there is an urgent need for real-life data about the safety and efficacy of mAbs and RMD among immunocompromised patients.

To fill this gap in knowledge we retrospectively assessed the safety and clinical outcomes of early treatment with mAbs and/or RMD among subjects with either primary or secondary immunodeficiencies, a diagnosis of SARS-CoV-2 infection, and who received these treatments in the Infectious Diseases outpatient clinic of Foundation IRCCS Ca’ Granda Ospedale Maggiore Policlinico located in Milan, Italy.

## 2. Materials and Methods

### 2.1. Patients and Study Design

We conducted a retrospective, analytic, cohort study focused on subjects with SARS-CoV-2 infection, diagnosed through a positive SARS-CoV-2 PCR on a nasopharyngeal swab, attending the Infectious Diseases outpatient clinic of Foundation IRCCS Ca’ Granda Ospedale Maggiore Policlinico, Milan, Italy. All patients received early treatment with mAbs (casirivimab/imdevimab, sotrovimab, or bamlanivimab/etesevimab) and/or RMD according to the Italian National Health System indications (Appendix A) from 25 November 2021 to 25 January 2022. Patients provided informed consent with regard to data processing.

Participants were classified as either belonging to the nonimmunocompromised or immunocompromised cohorts based on the absence or presence, respectively, of: (i) a history of any connective tissue disease, autoimmune disease, or primary immunodeficiency; (ii) a history of an active solid or hematologic tumour; (iii) neutropenia due to haematological cancer; (iv) a diagnosis of human immunodeficiency virus (HIV) infection or acquired immunodeficiency syndrome (AIDS); (v) a history of splenectomy, SOT, and/or HSCT; (vi) ongoing treatment with steroids (for at least 4 weeks), chemotherapy, and/or immunosuppressive agents.

Electronic medical records were reviewed, and anonymized data were abstracted on standardized data collection forms. A telephonic interview was conducted at least 28 days after treatment(s) administration to collect outcome variables through a specific questionnaire. Demographic data included sex, age, body mass index (BMI), and race. Clinical data included previous history of SARS-CoV-2 infection, COVID-19 vaccination, comorbidities, KDIGO classification, presence of primary or secondary immunodeficiencies, symptoms displayed, treatment administered, adverse drug reactions (ADRs) manifested, and outcomes reported. The study was approved by our Institutional Review Board (Milano Area 2, #328_2022bis, 26 April 2022) and was conducted in accordance with the Helsinki Declaration.

The primary outcome was to determine the frequency of ADRs among immunocompromised patients compared to immunocompetent patients. ADRs were graded according to the Common Terminology Criteria for Adverse Events (CTACAE) v.4.0. Secondary outcomes were to determine (i) the duration of molecular swab positivity and extent of symptoms, (ii) admission to the emergency department (ED) for any reason including COVID-19, (iii) hospitalization for any reason including COVID-19, (iv) admission to the intensive care unit (ICU) for any reason including COVID-19, (v) and death for any reason including COVID-19. SARS-CoV-2 PCR tests using nasopharyngeal swabs were employed to assess each patient’s clearance of infection. Admission and death for COVID-19 were defined according to the diagnosis reported on medical reports or death certificates. All the information collected for the follow-up was censored after 14 days from the last treatment administration.

### 2.2. Virologic Analyses

SARS-CoV-2 RNA detection on nasopharyngeal swabs and variant assessments were performed as previously described [13].

### 2.3. Statistical Analyses

Quantitative variables were described as median and interquartile ranges (IQR). Categorical variables were presented as frequencies and percentages. Chi-square tests were employed to compare frequencies of adverse events, hospitalization, and death between immunocompetent and immunocompromised subjects. Categorical variables were compared by using Pearson’s chi-square test or Fisher’s exact test, whereas continuous variables were compared by employing the Wilcoxon rank–sum test. The Kaplan–Meier plots were produced to compare the time to nasopharyngeal swab negativity and the time to the resolution of COVID-19 clinical manifestation between the two populations. The analyses were performed with R version 4.1.2.

## 3. Results

In this study 143 patients were included, and their baseline characteristics are reported in Table 1 and Appendix A. Most of the participants were immunocompromised (106/143, 74.1%, *p* < 0.001). Immunocompromised patients were younger (35/106 vs. 32/37 over 65 years old, *p* < 0.001) with a lower BMI (median 24.0 vs. 25.0 Kg/m^2^, *p* = 0.04) than the immunocompetent patients. Regarding immunocompromised-status definition, the most common classifying conditions were, being a kidney transplant recipient (23/143, 16.1%) and suffering from primary immunodeficiency (22/143, 15.4%) (Table 2).

Only 8/106 (7.5%) and 1/37 (4.8%) patients were not vaccinated among the immunocompromised and nonimmunocompromised patients, respectively. A primary vaccine series plus a booster dose at treatment start was already completed by 67/106 (63.2%) and 14/37 (67%) immunocompromised and nonimmunocompromised patients, respectively. 

Among those participants for whom the sequencing of the viral strain was performed (93/143), the vast majority were infected by the omicron (B.1.1.529) variant (88/93, 95%), only a minority were infected by the delta (B.1.617.2) variant (5/93, 5%). All patients declared signs and/or symptoms of COVID-19, among them the most frequently reported were fever (78/143, 55%) and cough (75/143, 52%). The full list of COVID-19 clinical manifestations reported by our patients is described in Appendix A.

The most frequently administered treatments were mAbs (122/143, 84.7%), and mainly sotrovimab (83/143, 58%), whereas RMD was employed only in 23 patients (16%). Two participants received both treatment options (Appendix A and Figure 1).

Twenty-three participants reported having experienced ADRs—one of which experienced both early and late ADRs for a total of 24 ADRs—without statistically significant differences between immunocompromised and immunocompetent patients (Table 3).

Only one grade-1 ADR (i.e., vertigo during infusion with complete remission upon discontinuation of the drug) was observed in an immunocompromised patient early in the treatment with mAbs (i.e., sotrovimab). No anaphylactic-type reactions were observed. Among late ADRs, the most commonly reported were the occurrence of fever (3/143, 2.1%) and skin rash (3/143, 2.1%) (Appendix A).

The median time between the onset of COVID-19 signs and/or symptoms and treatment administration was 6 (IQR 4–7) days. Among the immunocompromised patients, 11 were treated more than 7 days from the beginning of signs and/or symptoms of COVID-19. The median time from start of treatment to nasopharyngeal swab negativity and resolution of COVID-19 clinical manifestations were 10 (IQR 6–16) and 2.5 (IQR 1.0–6.0) days, respectively. No significant differences were observed between immunocompromised and immunocompetent patients (Appendix A).

Within the 14 days after treatment administration, eight participants (5.8%) all belonging to the immunocompromised group, had access to the emergency department. Five of them were hospitalized due to COVID-19. One patient (0.7%), an immunocompromised person who had received sotrovimab, died due to COVID-19 (Table 4).

## 4. Discussion

Our study provides evidence about the good safety and efficacy profiles of early treatment with RMD or mAbs among immunocompromised patients affected by mild/moderate COVID-19, mainly infected by the omicron (B.1.1.529) variant. Overall, only a minority of those treated reported ADRs; of those reports, all were of low severity, mainly with late appearance, and with no cases of anaphylaxis. No differences in clinical outcomes were observed between immunocompromised and immunocompetent patients, even though the first group displayed a trend toward a higher proportion of hospitalizations due to COVID-19. 

Regarding the ADRs, RMD has been in use for almost two years, and a large bulk of data has accumulated highlighting how its use can be associated mainly with an increase in serum creatinine (>10% of those treated), skin rash (<2% of those treated), and nausea (3–7% of those treated) [14]. As expected in a real-life study, we experienced a higher frequency of reported ADRs (4/22, 18%) compared to registration studies despite the reduced duration of early treatment (three vs. five days) compared with standard treatment, which should have reduced the risk of ADRs occurring. Nonetheless, it is reassuring that all the ADRs were mild and did not hamper the administration of the treatment. MAbs, by contrast, have been introduced more recently so there have been fewer opportunities to learn about the possible ADRs. In our study the most frequently employed mAb was sotrovimab, representing almost 60% of all the early treatments administered. We observed a frequency of adverse events among those receiving mAbs (20/122, 16.4%) which was quite similar to the frequencies (17–22%) reported in the COMET-ICE trial, which assessed efficacy and ADRs of sotrovimab in preventing the progression of mild to moderate COVID-19 to severe disease status [15]. Encouragingly, neither we nor the authors of the COMET-ICE trial reported any cases of anaphylaxis, a feared ADR when administering monoclonal antibodies. 

Interestingly, the proportion of hospitalizations in our study (8/143, 5.8%) was higher than those reported in randomized clinical trials. The COMET-ICE study [10], reported an all-cause hospitalization rate at 29 days after randomization of 1% (3 out of 291) in patients treated with sotrovimab. The phase 3 BLAZE-1 study reported a 29 day COVID-19-related hospitalization or death from any cause rate of 2.3% (12 out of 518 patients) in patients who received bamlanivimab/etesevimab [16]. The PINETREE study reported an all-cause hospitalization rate at 28 days of 1.8% (5 out of 279 patients) among patients treated with RMD [17]. The reason for these higher proportions of subjects hospitalized in our experience could be attributed to the inclusion of high-risk patients in our cohort and the fact that immunocompromised patients, which in our cohort represent the majority of those hospitalized, are underrepresented or even excluded from RCTs. For example, the BLAZE-1 study included a very low number of participants with either pre-existing immunologic conditions or concomitant treatment with immunosuppressive agents (1.5% and 4.9%, respectively). Further information on the use of early treatment for COVID-19 among immunocompromised patients is provided by other single-centre observational studies and case series, mostly regarding the administration of bamlanivimab, bamlanivimab/etesevimab, and casirivimab/imdevimab [18,19,20,21,22]. Despite the small number of patients enrolled and the inherent biases, these studies contributed to corroborating the safety profile and the therapeutic role of mAbs in nonhospitalized immunocompromised patients. In all these studies the proportion of subjects hospitalized exceeds the one observed in our cohort study; but similarly, no cases of anaphylaxis and absent or very uncommon early and late ADRs were reported.

Notably, despite many case reports describing the prolonged viral shedding occurring among immunocompromised patients [23,24,25,26], no difference in time from treatment start to SARS-CoV-2 nasopharyngeal swab negativity was observed between the two groups in our analysis. This might suggest that early treatment plays a role in helping immunocompromised subjects to clear the virus from the nasopharynx in a quicker way. If this is proved to be the case, it might be instrumental for individuals with comorbidities whose treatments (e.g., chemotherapies, surgery) might have been halted, or temporarily modified, because of COVID-19. Moreover, long-lasting viral replication among immunocompromised patients has been suggested as a driver of viral evolution and variant selection. Reducing the amount of time in which the virus can evolve in this population through the use of early treatments could then be useful from a wider, epidemiological perspective [27].

Our study also provides an insight into the outcomes of previously vaccinated immunocompromised patients treated with RMD or mAbs, as just 8/107 (7.5%) of them were not vaccinated. This information is lacking in RCTs which considered patients who had received any SARS-CoV-2 vaccine ineligible [10,17] or did not report data about the immunization status [16]. Additionally, observational studies concerning the use of mAbs in immunocompromised patients often included SOT recipients and haematological patients who had never been vaccinated [18,21]. Our study does not allow us to draw definite conclusions about the effectiveness of mAbs administration among vaccinated immunocompromised patients; however, it is reassuring that treatment with these drugs seems safe in those who could have antispike antibodies already circulating, as induced by immunization. 

We recognize that our study has some limitations, such as the retrospective design, the relatively small sample size, and the short follow-up period. It only involved outpatients selected according to Italian Medicines Agency (AIFA) regulations. This limits its generalizability as the criteria required for the dispensation of these treatments might differ in other countries. Moreover, a selection bias could be present as it is possible that not all subjects who met the criteria for receiving early treatment were informed of this opportunity by the general practitioner. Similarly, it is possible that not all subjects eligible for treatment were able to access our outpatient clinic, either because of logistical reasons, socioeconomic reasons, or because they had a performance status that did not allow access to clinical structures. Finally, a recall bias could have occurred, considering how information about late ADRs was collected through a telephonic questionnaire administered at least 28 days after the administration of treatment(s). 

In conclusion, to the best of our knowledge, this is the first study reporting the safety profile and clinical outcomes of the early treatment of COVID-19 in immunocompromised patients infected with the omicron (B.1.1.529) variant. Our results shed light on the feasibility and efficacy of early treatment in such a fragile population that is usually excluded from randomized clinical trials. However, further studies—also including molnupiravir, nirmatrelvir/ritonavir, and tixagevimab/cilgavimab—are very much needed to prove the rationale of their use and refine their administration in immunocompromised patients.

## Figures and Tables

**Figure 1 biomedicines-10-02002-f001:**
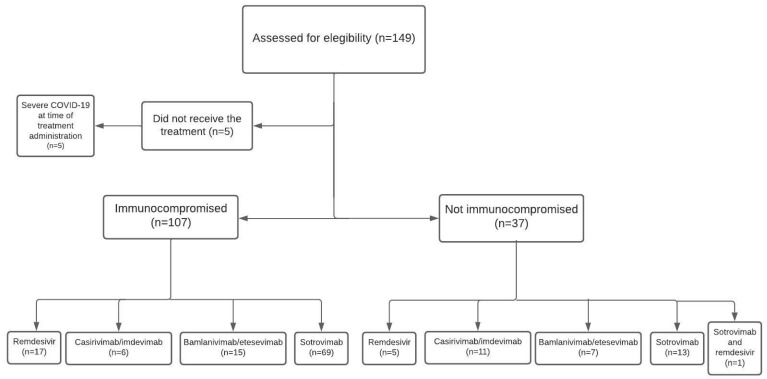
Study flowchart.

**Table 1 biomedicines-10-02002-t001:** Demographic and virologic characteristics of outpatients by subjects’ immune status.

			Immunocompromised		
	N	Overall, N = 143 *	No, N = 37 *	Yes, N = 106 *	*p*-Value ^†^
**Age** (years)	143				<0.001
20–64		76 (53%)	5 (14%)	71 (67%)	
65+		67 (47%)	32 (86%)	35 (33%)	
**Gender**	143				0.451
M		81 (57%)	19 (51%)	62 (58%)	
F		62 (43%)	18 (49%)	44 (42%)	
**Ethnic group**	143				0.603
Caucasian		136 (95%)	37 (100%)	99 (93%)	
African		3 (2.1%)	0 (0%)	3 (2.8%)	
Asian		3 (2.1%)	0 (0%)	3 (2.8%)	
Hispanic		1 (0.7%)	0 (0%)	1 (0.9%)	
**Vaccine dose** (s)	140				0.607
0		9 (6.4%)	1 (4.8%)	8 (6.7%)	
1		4 (2.9%)	1 (4.8%)	3 (2.5%)	
2		46 (33%)	5 (24%)	41 (34%)	
3		81 (58%)	14 (67%)	67 (56%)	
**BMI** (Kg/m^2^)	138	24.0 (22.0, 26.0)	25.0 (23.0, 28.0)	24.0 (21.0, 26.0)	0.032
**Time from last vaccine dose to COVID-19** (days)	131	85 (49, 162)	115 (63, 174)	80 (47, 154)	0.248
**SARS-CoV-2 variant**	93				>0.999
Omicron		88 (95%)	14 (93%)	74 (95%)	
Delta		5 (5%)	1 (7%)	4 (5%)	
**Previous SARS-CoV-2 infection**	142	8 (5.6%)	1 (2.7%)	7 (6.7%)	0.680

* Median (IQR) or Frequency (%) ^†^ Pearson’s Chi-squared test; Fisher’s exact test; Wilcoxon rank–sum test. Abbreviations: BMI = body mass index.

**Table 2 biomedicines-10-02002-t002:** Frequency of the conditions defining the immunocompromised status.

Immunosuppressive Condition	N = 143 *
**Connective tissue disease**	11 (7.7%)
**Solid tumour**	8 (5.6%)
Local	5 (3.5%)
Metastatic	3 (2.1%)
**Leukaemia**	7 (4.9%)
**Lymphoma**	13 (9.1%)
**HIV infection**	2 (1.4%)
**AIDS**	0 (0%)
**Splenectomy**	2 (1.4%)
**Neutropenia**	1 (0.7%)
**Primary immunodeficiency**	22 (15.4%)
**Autoimmune disease**	13 (9.1%)
**Bone marrow transplant recipients**	6 (4.2%)
Autologous	5 (3.5%)
Allogenic	1 (0.7%)
**Solid-organ transplant recipients**	42 (29.4%)
Kidney	23 (16.1%)
Liver	12 (8.5%)
Lungs	5 (3.5%)
**Long term steroid therapy ^†^**	51 (35.7%)
<20 mg/day	48 (33.6%)
≥20 mg/day	3 (2.1%)
**Biological immunosuppressor**	
Anti TNF-alfa	3 (2.1%)
Anti IL6	1 (0.7%)
Anti CD20	4 (2.8%)
Other(s)	10 (7.0%)
**Ongoing chemotherapy**	6 (4.2%)
**Ongoing antirejection therapy**	44 (30.8%)

* n (%) ^†^ Expressed as mg/day of prednisone. Abbreviations: AIDS = acquired immunodeficiency syndrome, HIV = human immunodeficiency virus.

**Table 3 biomedicines-10-02002-t003:** Adverse drug reactions according to immune status for those receiving remdesivir (**a**) and monoclonal antibodies (**b**).

(**a**)					
			**Immunocompromised**		
**Remdesivir**	**N**	**Overall, N = 23 ***	**No, N = 6 ***	**Yes, N = 17 ***	***p*-Value ^†^**
**Early ADRs**	22	0 (0%)	0 (0%)	0 (0%)	>0.999
**Late ADRs**	22	4 (18%)	1 (17%)	3 (19%)	>0.999
**ADRs grade**	4				
Grade 1		4 (100%)	1 (100%)	3 (100%)	
(**b**)					
			**Immunocompromised**		
**mAbs**	**N**	**Overall, N = 122 ***	**No, N = 32 ***	**Yes, N = 90 ***	***p*-Value ^†^**
**Early ADRs**	122	1 (0.8%)	0 (0%)	1 (1.1%)	>0.999
** ADRs grade**	1				
Grade 1		1 (100%)	0 (0%)	1 (100%)	
**Late ADRs**	119	19 (16%)	5 (16%)	14 (16%)	>0.999
** ADRs grade**	19				
Grade 1		18 (95%)	5 (100%)	13 (93%)	
Grade 5		1 (5.3%)	0 (0%)	1 (7.1%)	

* Median (IQR) or Frequency (%) ^†^ Fisher’s exact test. Abbreviations: ADRs = adverse drug reactions; mAbs = monoclonal antibodies.

**Table 4 biomedicines-10-02002-t004:** Treatment induced adverse events by therapy administered and stratified according to immunocompromised status.

			Immunocompromised		
	N	Overall, N = 143 *	No, N = 37 *	Yes, N = 106 *	*p*-Value ^†^
**Emergency department admission within 14 days from infusion**	138	6 (4.3%)	1 (2.8%)	5 (4.9%)	>0.999
COVID-19 related		5	0	5	
**Hospital admission within 14 days from infusion**	139	8 (5.8%)	0 (0%)	8 (7.8%)	0.109
COVID-19 related		5	0	5	
**ICU admission within 14 days from infusion**	138	0 (0%)	0 (0%)	0 (0%)	
**Death within 14 days from infusion**	138	1 (0.7%)	0 (0%)	1 (1.0%)	>0.999
COVID-19 related		1	0	1	

* Median (IQR) or Frequency (%) ^†^ Fisher’s exact test.

## Data Availability

Data will be provided on reasonable request.

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
