# Peer review of "Safety Profile and Outcomes of Early COVID-19 Treatments in Immunocompromised Patients: A Single-Centre Cohort Study"

_biomedicines, 2022, doi:10.3390/biomedicines10082002_

Round 1
Reviewer 1 Report
Thank you for giving me the opportunity to read and comment a report “Safety profile and outcomes of early COVID-19 treatments in immunocompromised patients: a single-centre cohort study”, by Biscarini S, et al.
In the reviewed manuscript, the safety and clinical outcomes of early treatment with mAbs and/or RMD among subjects with either primary or secondary immunodeficiencies and a diagnosis of SARS-CoV-2 infection has been investigated.
This paper is well written, correctly structured with a suitable research concept, the study limitations are addressed, and it is of relevance to readers of the journal. However, I include a comments for your consideration.
· In the subsection "Statistical analysis" it is indicated that quantitative variables were described as the mean or median. However throughout the manuscript the authors only use the median. Thus, it would be appropriate to eliminate the mean from the methods used.
· Tables 1, 3 and 4 indicate the use of statistical tests that are not described in the corresponding section, such as the Fisher's exact test and the Wilcoxon rank-sum test. It would be advisable for the authors to describe these tests in the subsection "statistical analysis".
· In Table 3 there are untranslated words in English, such as “die”.
Author Response
Reviewer 1
Thank you for giving me the opportunity to read and comment a report “Safety profile and outcomes of early COVID-19 treatments in immunocompromised patients: a single-centre cohort study”, by Biscarini S, et al.
In the reviewed manuscript, the safety and clinical outcomes of early treatment with mAbs and/or RMD among subjects with either primary or secondary immunodeficiencies and a diagnosis of SARS-CoV-2 infection has been investigated.
This paper is well written, correctly structured with a suitable research concept, the study limitations are addressed, and it is of relevance to readers of the journal. However, I include a comments for your consideration.
- In the subsection "Statistical analysis" it is indicated that quantitative variables were described as the mean or median. However throughout the manuscript the authors only use the median. Thus, it would be appropriate to eliminate the mean from the methods used.
Agreed and so modified.
- Tables 1, 3 and 4 indicate the use of statistical tests that are not described in the corresponding section, such as the Fisher's exact test and the Wilcoxon rank-sum test. It would be advisable for the authors to describe these tests in the subsection "statistical analysis".
We have described the above-mentioned tests at lines 123-125.
- In Table 3 there are untranslated words in English, such as “die”.
We have corrected the typos.
Reviewer 2 Report
The current retrospective study shows that remdesivir and monoclonal antibodies have minimal adverse drug reaction and favourable outcomes in immunocompromised patients. I have the following comments.
· In table 2, AIDS is listed as 0 (0%). Please double check this.
· In table 2, “mg/die” should be “mg/day”? please double check this.
· One concern would be the small sample size in table 3b, given the lower incidence of ADR.
· In table 4, the category name is suggested to be shorter.
· Regarding the hospitalization, early use of hydroxychloroquine at mild patients is hypothesized to be beneficial in reducing patients’ hospitalization (PMID: 33116907). Please elaborate on this in the discussion section.
Author Response
Reviewer 2
The current retrospective study shows that remdesivir and monoclonal antibodies have minimal adverse drug reaction and favourable outcomes in immunocompromised patients. I have the following comments.
- In table 2, AIDS is listed as 0 (0%). Please double check this.
We confirm that no patients with AIDS diagnosis have been enrolled in the study. Nonetheless, we verified the presence and reported this variable.
- In table 2, “mg/die” should be “mg/day”? please double check this.
We have corrected the typos.
- One concern would be the small sample size in table 3b, given the lower incidence of ADR.
We are fully aware of the limitations related to the small sample size, but we decided to report the data as collected to provide a full overview to the readership.
- In table 4, the category name is suggested to be shorter.
We decided to maintain these category labels to help the reader in understanding the table.
- Regarding the hospitalization, early use of hydroxychloroquine at mild patients is hypothesized to be beneficial in reducing patients’ hospitalization (PMID: 33116907). Please elaborate on this in the discussion section.
Considering the large bulk of studies (metanalyses, RCTs, cohort studies) showing the inefficacy of hydroxychloroquine in the prevention and treatment of COVID-19, we preferred not to mention this molecule in the discussion.
We sincerely hope that this revised version will now be acceptable for publication in Biomedicines.